# Clinical-Epidemiology of Tension-Type Headache among the Medical and Dental Undergraduates of King Khalid University, Abha, Saudi Arabia

**DOI:** 10.3390/jpm12122064

**Published:** 2022-12-14

**Authors:** Zia Ul Sabah, Shahid Aziz, Bayapa Reddy Narapureddy, Hassan Ahmed A. Alasiri, Hassan Yahya M. Asiri, Abdulkhaliq Hadi H. Asiri, Ahmad Abdullah H. Alsulami, Nawaf Khalid Ahmad Hassan, Shaik Mohammed Asif, Shmookh Mohsen Alsyd

**Affiliations:** 1Department of Medicine, College of Medicine, King Khalid University, Abha 62529, Saudi Arabia; 2Department of Public Health, College of Applied Medical Sciences, King Khalid University, Abha 62529, Saudi Arabia; 3College of Medicine, King Khalid University, Abha 62529, Saudi Arabia; 4Department of Diagnostic Science and oral Biology, College of Dentistry, King Khalid University, Abha 62529, Saudi Arabia; 5Public Health, Ministry of Health, Riyadh 12613, Saudi Arabia

**Keywords:** headache/epidemiology, etiological factor, triggers of TTH, medical students, prevalence, tension-type headache

## Abstract

**Introduction (background)**: Headache is the primary complaint among students. Headaches mostly have multifactorial causes. The degree of headache severity significantly impacts attitudes, behavior, and academic performance. **Objectives**: Here, we investigate the demographic epidemiology of tension-type headaches (T.T.H.), and determine the clinical presentation and triggers of the tension-type among headache sufferers. **Methods (settings, design):** An institutional-based cross-sectional study (descriptive) was conducted on the medical and dental undergraduates at King Khalid University, Abha, Saudi Arabia, from 1 July 2021 to 31 December 2021. Data were gathered using a pre-designed questionnaire. A consecutive sampling method was used in a COVID-19-constrained environment. After preliminary screening of the study population, 460 samples were included. An electronic questionnaire was shared with them, and they were requested to respond. **Results:** More than half of the participants (258, 56.1%) experienced tension-type headaches, while the remaining 202 (43.9%) never felt a headache. Tension-type headaches manifested as heaviness of the head (44, 17.0%), tightness (126, 48.8%), and dull aching pain (66, 25.7%). **Conclusions:** T.T.H. is a prevalent condition with a significant impingement on academic work, and psychological health. Tension-type headache sufferers are advised to keep daily diaries to determine triggers, and plan for prevention and treatment progression.

## 1. Introduction

A headache is a pain in the head; pain may be generalized so that the pain can be sensed all over the head. For some individuals, it may start in one area of the head [1]. Globally, almost half of the adult population suffered headaches at least once within the last year; headache attacks are categorized into primary and secondary headaches. Secondary headaches are associated with underlying medical conditions, and primary headaches include clusters, migraine, and T.T.H.s [2]. The International Classification of Headache Diagnosis I (ICHD I) coined the term tension-type-headache (T.T.H.) [3,4]. T.T.H. has previously been stated with several other terms (e.g., simple headache, stress headache, psychogenic headache, muscle contraction headache). Currently, the term tension-type-headache (T.T.H.) is commonly used [5]. A headache in T.T.H. is characterized by “tightening” and “pressing” pain around the head and can occur with various duration and frequencies. T.T.H. pain involves contracting muscles in the neck, jaw, face, and scalp [6]. T.T.H.s may also be caused by jaw clenching, depression, anxiety, or insomnia. Some conditions, such as temporomandibular joint pain and fibromyalgia with idiopathic oro-facial pain, mimic tension-type headaches and need to be differentiated before initiating treatment [7]. Unfortunately, the T.T.H. mechanism remains unclear; tension-type headache is a multifactorial disorder with several pathophysiological mechanisms. However, cluster headaches and migraines are thought to be initiated because of neurological dysfunction involving the cranial vessels [5]. The International Headache Society classified the T.T.H. based on the frequency of the attacks classified into episodic and chronic types [3,8]. Some concepts about the pathophysiology of episodic T.T.H. (headache 1–14 days per month) due to peripheral mechanisms (myofascial nociception), and chronic T.T.H. (headache 15 days per month) due to central mechanisms (sensitisation and inadequate endogenous pain control) [9,10].

Globally, three billion people are suffering from migraines or T.T.H.s. They are among the ten most common causes of burden worldwide, after dental caries and latent tuberculosis infection; T.T.H. is the third most prevalent disorder globally [11]. Globally, the lifetime prevalence of T.T.H. is 78% in the general adult population [12]. More than half (56%) of the adult population suffers from headache disorders; nearly 42% of people suffer from T.T.H.s, 11% from migraines, and 3% from chronic daily headaches [13,14]. T.T.H. is the most common primary headache among the general population. The prevalence of T.T.H.s reported by different studies ranges from 30% to 78% [2,15]. T.T.H. was reported more frequently among females than males, and the female-to-male ratio of T.T.H. was 1.8:1 [16]. The headache prevalence in Saudi Arabia ranged from 8% to 12%; T.T.H.s varied from 3.1% to 9.5%, among these headaches [17]. The headache prevalence among medical students was 53.78%, and for tension-type headache it was 41.66%. For migraine headache the prevalence was 7.1% [18]. Mohammed Jumah et al. revealed in K.S.A. that 1-year prevalence for all types of headaches was 77.2%. The prevalence of headache types, according to adjusted 1-year prevalence, were 34.1% for T.T.H. and 25.0% for migraine [19].

Headache is a significant common problem among university students, compared to other populations. Many studies have reported that headaches are more prevalent among the student community [20,21]. Headache is one of the significant complaints among medical students, perhaps due to the challenging syllabus, multiple exams, sleep deprivation, family stresses, long working hours, and psychological issues [1,2,22,23]. Stress is common in modern daily life, resulting from rapid environmental changes. In the present competitive world, students need to perform multiple tasks, and the increased stresses on students are why the student age is called the age of stress. Students were exposed to academic pressure from academic tests, assignments, attendance, and other university requirements; these stressors sometimes exceed their ability to cope [6]. Among medical students, the most reported headaches are T.T.H.s and migraines; these might be due to the long working hours, leading to fatigue, stress, and anxiety. Students with frequent headaches mostly suffer from lost study days which lead to reduced academic performance, poor concentration, and poor attitudes and behavior [24].

The current study has planned to identify the demography and epidemiology of tension-type headaches among undergraduate medical students to determine the clinical presentation and triggers of T.T.H. among the study participants, and assess the attitude towards T.T.H. management.

## 2. Methodology

Study on the medical and dental undergraduate students at King Khalid University, Abha, Saudi Arabia. An institutional-based cross-sectional (descriptive) study was conducted in a COVID-19-constrained environment from 1 July 2021 to 31 December 2021. The data were gathered using a pre-designed survey questionnaire. The researchers developed the questionnaire, and a panel of three experts independently validated the questionnaire. The questionnaire comprises three parts: Part 1 consists of socio-demographic data, such as gender, age, college, and marital status. Part 2 discusses the clinical characteristics, onset, aggravating and relieving factors of T.T.H.s. Part 3 examines the effect of T.T.H.s on students′ academic performance, attitudes, and behavior. The study participants were recruited after preliminary screening telephonically by expert (qualified medical practitioners) authors. The participants excluded those with nasal deformity, ear problems, chronic psychological conditions, individuals with visual disorders who were not willing to participate and were on long leave.

The study participants met the inclusion criteria, and the self-administered electronic study tool (Google Forms) link was shared with them through WhatsApp and emails. Before beginning to fill out the google form, this study′s objectives and goals were provided for participants, in the local language and English also. At the beginning of the survey, the tool explained data confidentiality and assured participants that personal data, that can recognize the individuals, were not collected; the data were stored in coded form only, after assuring them digital consent was obtained; and participants accepted the privacy policy for the protection of personal data, before completing the survey. A consecutive sampling technique was used to recruit the study sample. The minimum sample size was determined using the formula 4pq/d^2^ with a 95% confidence interval of 452. This study ultimately included 460 participants.

### 2.1. Ethical Issues

As per the Declaration of Helsinki, digital consent was obtained from the participants before the study proceeded. Institutional Ethical approval (E.C.M. #2021-4909) was acquired from the Research Ethics Committee (HAPO-06-B-001) of King Khalid University, Abha, Saudi Arabia, K.S.A.

### 2.2. Statistical Analysis

The validated data were analyzed using Statistical Package for the Social Sciences (IBM SPSS Statistics) for Windows, version 21 (I.B.M. Corp., Armonk, NY, USA). All the collected data were downloaded in a Microsoft Office 2019 Excel spreadsheet from Google Forms. Categorical variables, such as demographic data, academic data, headache-related data, clinical features, and aggravating and relieving factors were expressed in proportions to test the hypothesis. The Chi-square test was applied. The *p*-value of <0.05 was considered significant at the 95% confidence interval level.

## 3. Results

A total of 460 students submitted the completed study questionnaire; 256 (55.7%) belonged to the college of medicine, and 204 (44.3%) belonged to the college of dentistry, respectively. The study sample is slightly higher among males (261, 56.7%) than females (199, 43.3%), with a ratio of 1.3:1. Among the 460 participants, the participants had a mean age of 22.3 ± 4.9 years (ranging from 18 to 27 years) (Table 1).

T.T.H. was present in 56.1% (258), while 43.9% (202) had not experienced T.T.H. attacks; out of 258 tension-type headaches, 118 (45.2%) were males, and 140 (70.3%) were females, with a ratio of female to male 1.18:1. Out of 204 dental undergraduate students, the majority (128, 62.7%) of them reported T.T.H. attacks, compared to (130, 50.7%) of the medical college undergraduate participants; the difference was of statistical significance (*p* < 0.01). The female gender (140, 70.4%) experienced more attacks of T.T.H., compared to the male participants (118, 45.2%); the difference was statistically significant (*p* < 0.001). T.T.H. was reported in higher rates among married (7, 77.8%), senior students, and those in their clinical years of study. Details have been provided in Table 2.

Headache was felt as a heaviness/fullness of the head among 44 (17%) students, a sensation of a tight band around the head among 126 (48.7%), numbness was felt by 16 (6.2%), and dull aching pain by 62 (24%). The onset of headache was reported as sudden onset (52, 20.1%) and gradual (54, 20.9%), but a majority of the 142 (55%) students had no specific onset (variable). Headaches appeared in the morning (30, 11.6%); some felt headaches more frequently at the end of the day (47, 18.2%). The headache attack lasted for hours among most students (179, 69.3%), and was mainly bilateral (191, 74.1%) (Table 3).

More than half of the students complained of T.T.H. attacks that started after puberty (241, 93.5%). The majority of participants 329 (71.5%) did not know their family members′ T.T.H. status. Family history of T.T.H. was reported among 62 (13.5%) students, and family history was strongly associated with the T.T.H., and the difference was statistically significant (*p* < 0.001). Among the 51 (11.0%) smokers, more than half of the smokers were T.T.H. sufferers (32, 62.7%). Out of 140 female participants, 29 (11.2%) reported that headache was aggravated by their menstrual cycle. Additionally, T.T.H. was more associated with sleep deprivation and oversleeping; sleep of lesser duration (<5 h) and more than 8 h sleep aggravated the T.T.H. episodes, compared to adequate sleep of 6–8 h per day. The difference was statistically significant (0.001). Most of the study participants (376, 81.7%) had adequate sleep (6–8 h), among the T.T.H. sufferers. Mood swings (147, 56.9%) was the most common aura for the T.T.H., followed by neck pain (21, 8.1%), dizziness (19, 7.3%), tingling sensation (18, 6.9%), blurring of vision/photophobia (5, 1.9%), and fatigue activity performed with effort (14, 5.4%). Most of the participants (189, 73.2%) expressed that psychological stress was the most common headache trigger, followed by physical effort (41, 15.8%), drinking too much caffeine (9, 3.5%), and hunger (6, 2.3%). Most often (161, 62.4%), the headaches could be relieved by resting, such as lying down/sleeping in a quiet, dark room. The details of etiological factors for T.T.H.s are provided in Table 4.

A few participants (46, 17.8%) needed medical attention, and 186 (72.1%) reported that headaches interfered with their daily activities. Many (168, 65.1%) used analgesics to get rid of headaches (Table 4), and 19 (76%) of the undergraduates, who were suffering from T.T.H., purchased the drugs over the counter without any consultation with a physician. (Table 5)

## 4. Discussion

The current study has been conducted to establish the epidemiological factors contributing to T.T.H. among the student community. A total of 55.7% belonged to the medical college, and 44.3% belonged to the dental college. Out of 460 participants, nearly 56.7% were males, and 43.3% were females. The sample distribution was statistically significant (*p* < 0.001). The majority of the participants belonged to the 20–25 years age bracket. This is the usual age for undergraduate study. The majority of the participants were unmarried. Only a few were married, and those married were among the females. The study by M.S. alkarrash et al., also noted concordant results [25].

The overall prevalence of T.T.H. was 56.1%, and this was greater among females (70.3%) than males (45.2%) with a ratio of 1.18:1, and the difference was statistically significant (*p* < 0.05). Similar studies conducted by Stovner et al. [13] and Muayqil et al. [15] have reported comparable findings: the female-to-male ratio is about 5:4, and 1.8:1. Individuals in the higher age group and with a higher education level are more at risk for getting a tension-type headache. Another study in Saudi Arabia by Mohammed et al. [19] aligned with this study′s results. T.T.H. was the most common type of headache (42.9%), followed by migraine at 25% [19]. Headaches were predominantly in the female population. In another similar study conducted in Riyadh, Saudi Arabia, by Almesned et al. [18], the prevalence of T.T.H.s was 41.66%, with almost equal distribution of headaches in males (42%) and females (41.2%). The married students complained more about T.T.H. than the unmarried; this may be due to family stress and academic stress. Dental program students had T.T.H. more than the medical students. Whereas other studies conducted by MS Alakarrash et.al., in Syria [25] and Ruba M Saeed in Madina K.S.A. [26], observed that dental college students were having less tension-type headaches than medical undergraduates. This high prevalence in dental students in this study might be due to the COVID-19 pandemic; the dental college students were more worried about the COVID-19 infection spreading e.g., through the saliva, and contact with other mouth secretions [27].

Regarding the presentation of clinical features of T.T.H., tightness of the head is the most typical symptom, followed by heaviness or fullness, such as a tight band around the head. The onset of headache attacks was variable (sudden or gradual), and lasted for hours among three-quarters of the students. Additionally, T.T.H. attacks were bilateral among three out of four participants. Similar studies conducted by Pellegrino et al. [28], Fernandez et al. [29], and Hassan M et al. [6] also reported, almost coinciding findings. T.T.H.s start in the daytime and increase slowly throughout the day for most students [6].

More than half of the students complained of T.T.H. attacks that started after puberty (93.5%). The aura (prodrome) in the current study showed that more than half of the students reported experiencing mood change before the headache attack, one-tenth had neck pain, and about one-tenth experienced easy fatigability with low effort. The same type of diurnal variation of T.T.H. was observed by various other studies conducted in different locations: Mona Hassan et al. [6], Schytz et al. [30], and Spierings et al. [31]. This study noted that menstruation among female students was associated with frequent T.T.H. (catamenial) attacks [32]. The Spierings et al. [31] study also noticed that menstruation is one of the triggers for tension-type headaches. The most commonly reported trigger factors were fatigue and stress. Other factors included physical activity, sleep deprivation, hunger, and having too much coffee. Studies conducted by the European Headache Federation et al. [33], another study in Saudi Arabia by Ghada Kamal Gouhar [21], and Al Jumah et al. [19] also reported parallel triggers for T.T.H. and migraine. Most participants reported that the T.T.H. attacks were relieved by resting (lying down and sleeping) in a calm, dark room. Other relieving factors included keeping active in their activity, head massages, and cold packs applied to their eyes, head, and neck. A study by Palacios-Ceña M et al. [34] also observed consistency with this study regarding relieving factors to relieve the tension-type headache.

Regarding the effect of headaches on students′ attitudes, behavior, and daily activities, this study revealed that only 17.8% of the students with T.T.H. needed formal medical care; however, two-thirds of these students used analgesics to relieve their pain and discomfort. Similar studies, conducted by different authors, had findings that were also in concordance with this study [25,35]. Two-thirds of the students reported that headaches disrupted their activities of daily living. The results of Al Hasahel et al.′s [36] study in Kuwait among students align with this study′s findings regarding the effect of headaches on daily life activities [19].

## 5. Conclusions

Headaches are a frequent complaint among university undergraduate students. They commonly experience psychological stress and physical symptoms that affect their daily life activities. There is a substantial relationship between level of stress and T.T.H., with severity and duration of T.T.H. Some food items aggravate T.T.H., and T.T.H. can also be associated with the menstrual cycle. Sufferers should keep a daily record of the frequency, duration, time of initiation, aggravating factors, and severity of T.T.H. attacks, and also the relieving factors such as sleep, head massage, cold fomentation, and medications used. This would be helpful for T.T.H. sufferers, so they can avoid those triggers and plan for the prevention of T.T.H. Additionally, it also helps doctors to design the course of management for severe T.T.H. cases.

### Limitations

It is paramount to foster a stress-free natural world that will enable the students to put full effort into excelling in their academics. This study was conducted in a COVID-19 pandemic-restricted time. Selection bias is likely in the sample; it focuses on the medical communities, and neglects the other branches due to COVID constraints. This study has been carried out in one university with a limited sample; much of the data are collected from known cases. Therefore, results may not be extrapolated completely. A study should be conducted with a large sample, using a more systematic sampling method. This might improve the representativeness and generalizability of the findings, and a national policy could be developed to address this problem. Self-reporting is another limitation of the present study, with some participants perhaps giving socially accepted responses.

## Figures and Tables

**Table 1 jpm-12-02064-t001:** Distribution of study subjects based on demographic profile.

		Gender						*p* Value
		Female		Male		Total		
		Count	Row N %	Count	Row N %	Count	Row N %	
Age	18–19 Yrs.	11	64.7%	6	35.3%	17	100%	X^2^ = 4.49 Df: 2 *p* > 0.05 Not significant
	20–25 Yrs.	184	42.9%	245	57.1%	429	100%	
	25 Yrs.	4	28.6%	10	71.4%	14	100%	
Marital status	Married	7	77.8%	2	22.2%	9	100%	-
	Unmarried	192	42.6%	259	57.4%	451	100%	
Program	Dental	132	64.7%	72	35.3%	204	100%	X^2^ = 68.69 Df: 1 *p* < 0.0001 significant
	Medicine	67	26.2%	189	73.8%	256	100%	
Academic level	2nd level	23	62.2%	14	37.8%	37	100%	X^2^ = 74.2 Df: 9 *p* < 0.001 Significant
	3rd level	12	26.7%	33	73.3%	45	100%	
	4th level	48	81.4%	11	18.6%	59	100%	
	5th level	13	40.6%	19	59.4%	32	100%	
	6th level	34	63.0%	20	37.0%	54	100%	
	7th level	16	30.2%	37	69.8%	53	100%	
	8th level	28	34.6%	53	65.4%	81	100%	
	9th level	5	38.5%	8	61.5%	13	100%	
	10th level	11	22.9%	37	77.1%	48	100%	
	11th level	9	23.7%	29	76.3%	38	100%	
	Total	199	43.3%	261	56.7%	460	100%	

**Table 2 jpm-12-02064-t002:** Distribution of subjects based on demographic profile and T.T.H. history.

		T.T.H.						*p* Value
		No		Yes		Total		
		No	%	No	%	No	%	
Gender	Female	59	29.6%	140	70.4%	199	100.0%	X^2^ = 28.9 Df: 1 *p* < 0.001 Significant
	Male	143	54.8%	118	45.2%	261	100.0%	
Age	18–19 Yrs.	7	41.2%	10	58.8%	17	100.0%	X^2^ = 0.42 Df: 2 *p* >> 1.0 Not significant
	20–25 Yrs.	190	44.3%	239	55.7%	429	100.0%	
	25 Yrs.	5	35.7%	9	64.3%	14	100.0%	
Marital status	Married	2	22.2%	7	77.8%	9	100.0%	X^2^ = 1.7 Df: 1 *p* > 0.18 Not significant
	Unmarried	200	44.3%	251	55.7%	451	100.0%	
Program	Dental	76	37.3%	128	62.7%	204	100.0%	X^2^ = 6.5 Df: 1 *p* < 0.01 Significant
	Medicine	126	49.2%	130	50.8%	256	100.0%	
Academic level	10th level	29	60.4%	19	39.6%	48	100.0%	X^2^ = 26.0 Df: 9 *p* < 0.002 Significant
	11th level	16	42.1%	22	57.9%	38	100.0%	
	2nd level	19	51.4%	18	48.6%	37	100.0%	
	3rd level	26	57.8%	19	42.2%	45	100.0%	
	4th level	17	28.8%	42	71.2%	59	100.0%	
	5th level	7	21.9%	25	78.1%	32	100.0%	
	6th level	21	38.9%	33	61.1%	54	100.0%	
	7th level	19	35.8%	34	64.2%	53	100.0%	
	8th level	40	49.4%	41	50.6%	81	100.0%	
	9th level	8	61.5%	5	38.5%	13	100.0%	
	Total	202	43.9%	258	56.1%	460	100.0%	

**Table 3 jpm-12-02064-t003:** Distribution of study subjects based on Clinical features of tension headache.

		Gender											
		Female						Male					
		TTH						TTH					
		No		Yes		Total		No		Yes		Total	
		No	%	No	%	No	%	No	%	No	%	No	%
Nature of tension headache	Not Applicable	59	96.7%	2	3.3%	61	100.0%	143	94.7%	8	5.3%	151	100.0%
	Heaviness of head	0	0.0%	22	100.0%	22	100.0%	0	0.0%	22	100.0%	22	100.0%
	Numbness of the head	0	0.0%	7	100.0%	7	100.0%	0	0.0%	9	100.0%	9	100.0%
	Dull aching pain on the total head	0	0.0%	37	100.0%	37	100.0%	0	0.0%	25	100.0%	25	100.0%
	Tight banding pressure of the head	0	0.0%	72	100.0%	72	100.0%	0	0.0%	54	100.0%	54	100.0%
Onset of T.T.H.	Not Applicable	59	96.7%	2	3.3%	61	100.0%	143	94.7%	8	5.3%	151	100.0%
	Suddenly	0	0.0%	35	100.0%	35	100.0%	0	0.0%	17	100.0%	17	100.0%
	Gradually	0	0.0%	22	100.0%	22	100.0%	0	0.0%	32	100.0%	32	100.0%
	Varies time to time	0	0.0%	81	100.0%	81	100.0%	0	0.0%	61	100.0%	61	100.0%
Duration of Headache	Minutes	0	0.0%	15	100.0%	15	100.0%	0	0.0%	24	100.0%	24	100.0%
	Hours	0	0.0%	103	100.0%	103	100.0%	0	0.0%	76	100.0%	76	100.0%
	Days	0	0.0%	20	100.0%	20	100.0%	0	0.0%	10	100.0%	10	100.0%
	Not Applicable	59	96.7%	2	3.3%	61	100.0%	143	94.7%	8	5.3%	151	100.0%
On which side of the head feel the headache	Not Applicable	59	96.7%	2	3.3%	61	100.0%	143	94.7%	8	5.3%	151	100.0%
	Left side	0	0.0%	7	100.0%	7	100.0%	0	0.0%	8	100.0%	8	100.0%
	Right side	0	0.0%	21	100.0%	21	100.0%	0	0.0%	21	100.0%	21	100.0%
	Both sides	0	0.0%	110	100.0%	110	100.0%	0	0.0%	81	100.0%	81	100.0%
The pattern of the headache	More during day time	0	0.0%	14	100.0%	14	100.0%	0	0.0%	16	100.0%	16	100.0%
	More at the end of day	0	0.0%	25	100.0%	25	100.0%	0	0.0%	22	100.0%	22	100.0%
	Not specific	0	0.0%	99	100.0%	99	100.0%	0	0.0%	72	100.0%	72	100.0%
	Not Applicable	59	96.7%	2	3.3%	61	100.0%	143	94.7%	8	5.3%	151	100.0%
No of days had headaches in the past month	1–3 days	0	0.0%	44	100.0%	44	100.0%	0	0.0%	49	100.0%	49	100.0%
	4–6 days	0	0.0%	47	100.0%	47	100.0%	0	0.0%	30	100.0%	30	100.0%
	>6 days	0	0.0%	47	100.0%	47	100.0%	0	0.0%	31	100.0%	31	100.0%
	Not Applicable	59	96.7%	2	3.3%	61	100.0%	143	94.7%	8	5.3%	151	100.0%
How long does each headache attack last?	<2 h	0	0.0%	37	100.0%	37	100.0%	0	0.0%	44	100.0%	44	100.0%
	2–4 h	0	0.0%	45	100.0%	45	100.0%	0	0.0%	32	100.0%	32	100.0%
	4–24 h	0	0.0%	32	100.0%	32	100.0%	0	0.0%	18	100.0%	18	100.0%
	24–72 h	0	0.0%	9	100.0%	9	100.0%	0	0.0%	6	100.0%	6	100.0%
	Not Applicable	59	77.6%	17	22.4%	76	53.1%	143	88.8%	18	11.2%	161	100.0%
	Total	59	29.6%	140	70.4%	199	100.0%	143	54.8%	118	45.2%	261	100.0%

**Table 4 jpm-12-02064-t004:** Distribution of study subjects based on aggravating and relieving (triggers) factors vs. tension headache.

		T.T.H. History						*p* = Value
		No		Yes		Total		
		No	%	No	%	No	%	
T.T.H. started	Childhood	0	0.0%	9	100.0%	9	100.0%	-
	After Puberty	0	0.0%	241	100.0%	241	100.0%	
	Not applicable	202	96.2%	8	3.8%	210	100.0%	
Family history	Don’t know	197	59.90%	132	40.10%	329	100.00%	X^2^ = 119 Df: 2 *p* < 0.0001 Significant
	No	3	4.30%	66	95.70%	69	100.00%	
	Yes	2	3.20%	60	96.80%	62	100.00%	
Smoking	No	183	44.72%	226	55.3%	409	100.00%	X^2^ = 26.9 Df: 1 *p* < 0.0001 Significant
	Yes	19	37.25%	32	62.74%	51	100.00%	
Sleep duration	<5 h	2	3.40%	56	96.60%	58	100.00%	X^2^ = 64 Df: 2 *p* < 0.0001 Significant
	6–8 h	198	52.70%	178	47.30%	376	100.00%	
	>8 h	2	7.70%	24	92.30%	26	100.00%	
T.T.H. Menstrual association	Don’t know	0	0.0%	56	100.0%	56	100.0%	-
	No	0	0.0%	59	100.0%	59	100.0%	
	Yes	0	0.0%	29	100.0%	29	100.0%	
	Not Applicable	202	63.8%	114	36.2%	316	100.0%	
T.T.H. Severity	Mild T.T.H.	0	0.0%	51	100.0%	51	100.0%	
	Moderate	0	0.0%	138	100.0%	138	100.0%	
	Severe T.T.H.	0	0.0%	59	100.0%	59	100.0%	
	Not Applicable	202	95.3%	10	4.7%	212	100.0%	
Do you feel any of the following before the headache starts?	Mood changes	0	0.0%	147	100.0%	147	100.0%	-
	Neck pain	0	0.0%	21	100.0%	21	100.0%	
	Dizziness	0	0.0%	19	100.0%	19	100.0%	
	Tingling or numbness	0	0.0%	18	100.0%	18	100.0%	
	Daily activity with effort	0	0.0%	15	100.0%	15	100.0%	
	Change of Appetite	0	0.0%	14	100.0%	14	100.0%	
	Blurring of vision	0	0.0%	5	100.0%	5	100.0%	
	Irritable	0	0.0%	5	100.0%	5	100.0%	
	Zig-zag lines	0	0.0%	4	100.0%	4	100.0%	
	Not Applicable	202	95.3%	10	4.7%	212	100.0%	
Do any of the following cause headaches?	Psychological Stressors	0	0.0%	189	100.0%	189	100.0%	-
	Physical stress	0	0.0%	41	100.0%	41	100.0%	
	Drinking large amount of Caffeine	0	0.0%	9	100.0%	9	100.0%	
	Hunger	0	0.0%	6	100.0%	6	100.0%	
	Eating specific item	0	0.0%	3	100.0%	3	100.0%	
	Not Applicable	202	95.3%	10	4.7%	212	100.0%	
Relieving factor for your headache?	Sleep or lying down	0	0.0%	161	100.0%	161	100.0%	-
	Head massage	0	0.0%	62	100.0%	62	100.0%	
	Cold compresses on the head or neck.	0	0.0%	19	100.0%	19	100.0%	
	Physical activity and sports	0	0.0%	6	100.0%	6	100.0%	
	Not Applicable	202	95.3%	10	4.7%	212	100.0%	
Total		202	43.90%	258	56.10%	460	100.00%	

**Table 5 jpm-12-02064-t005:** Distribution of study sample based on attitudes, behavior, and c.

Consequences	No (258)	%
Needed medical care due to headache		
Yes	46	17.8%
No	212	82.2%
Headaches interfere daily activities		
Yes	186	72.1%
No	72	27.9%
How often do you need to stop your work or daily activities to deal with headache symptoms?		
None of the time	32	17.2%
Sometimes	105	56.5%
Most of the times	44	23.7%
All the time	5	2.7%
Do you use any illegal (sedative) drugs?		
Yes	3	1.2%
No	255	98.8%
Use analgesics for headache		
Yes	168	65.1%
No	90	34.9%
If using analgesics, did you have side effects?		
Yes	25	14.9%
No	143	85.1%
If using analgesics, Is it purchased over the counter? (n = 25)		
Yes	19	76%
No	6	24%

## Data Availability

Data available with Author’s sharing does not apply to this article.

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
