# Peer review of "Clinical-Epidemiology of Tension-Type Headache among the Medical and Dental Undergraduates of King Khalid University, Abha, Saudi Arabia"

_jpm, 2022, doi:10.3390/jpm12122064_

Round 1

Reviewer 1 Report

The authors conducted clinical epidemiology of tension-type headache (TTH) among undergraduate students. The data was collected from 460 students with a questionnaire designed to determine the clinical presentation of the TTH patients.

Generally, the manuscript is a standard epidemiology report with a well-designed questionnaire. Sample size are reasonable, and the statistical methods were correctly chosen. However, I found several sections of the manuscript are not well written, with an oversimplified conclusion from the results that need to be revised before further consideration. Here are some major issues:

1.      The introduction is not well written. The background of TTH is lengthy and introduces several concepts that were not discussed later. For example, the mechanism of TTH, the definition of episodic/chronic types. The paragraphs need reorganization, I suggest they should be reorganized as Background of TTH, Global epidemiology of TTH, then discuss the impact on students. There are several previously published works about the epidemiology of TTH. Authors should also discuss the deficiency of the prior researches and clearly address the reasoning of this research.

2.      The discussion described several selected statistical results in agrees with previous published studies in a separated manner. Previous research has concluded TTH is highly correlated with education level(Lipton, JAMA, 1998). Based on the results and selection of population is all medical students, I would assume stress and lack of rest is the main driver of the TTH. However, I feel the manuscript lacks a clearly summarized conclusion about the results.

There are some minor issues:

1.      Format of the reference is not uniform.

2.      Ref to ICHD-3 is not correct: International Classification of Headache Disorders-3 (beta version). Cephalalgia 2013;33:629–808.

3.      Date range of data is not consistent in the abstract and methodology

4.      Avoid using language like "similar things"

Author Response

Thank you so much for your kind words about our Research. We modified it as per your suggestion. Kindly review the article, and if any further needs to be improved, please give us a chance to modify it as per the journal guidelines. 

Open Review

English language and style

( ) English very difficult to understand/incomprehensible
( ) Extensive editing of the English language and style required
(x) Moderate English changes required
( ) English language and style are fine/minor spell check required
( ) I don't feel qualified to judge the English language and style

Reply: The English were edited by one of the English language professional if any further is required we will do it accordingly

Yes

Can be improved

Must be improved

Not applicable

Does the introduction provide sufficient background and include all relevant references?

( )

( )

(x)

( )

Are all the cited references relevant to the research?

( )

( )

(x)

( )

Is the research design appropriate?

(x)

( )

( )

( )

Are the methods adequately described?

(x)

( )

( )

( )

Are the results clearly presented?

( )

( )

(x)

( )

Are the conclusions supported by the results?

( )

(x)

( )

( )

Comments and Suggestions for Authors

Reply: The Introduction, results, conclusions, and the references have been revamped and aligned with the latest available evidence.

The authors conducted clinical epidemiology of tension-type headache (TTH) among undergraduate students. The data was collected from 460 students with a questionnaire designed to determine the clinical presentation of the TTH patients.

Reply: The study was conducted to determine the epidemiology of TTH and the symptoms and triggers for TTH to plan preventive strategies in further research.

Generally, the manuscript is a standard epidemiology report with a well-designed questionnaire. Sample sizes are reasonable, and the statistical methods were correctly chosen. However, I found several sections of the manuscript are not well written, with an oversimplified conclusion from the results that need to be revised before further consideration. Here are some major issues:

  1. The introduction is not well written. The background of TTH is lengthy and introduces several concepts that were not discussed later. For example, the mechanism of TTH, and the definition of episodic/chronic types. The paragraphs need reorganization. I suggest they should be reorganized as Background of TTH, Global epidemiology of TTH, then discuss the impact on students. There are several previously published works about the epidemiology of TTH. Authors should also discuss the deficiency of the prior research and address the reasoning behind this research.

Reply: Introduction has been rectified and modified as per the reviewers' suggestion

  1. The discussion described several selected statistical results in agrees with previous published studies in a separated manner. Previous research has concluded TTH is highly correlated with education level(Lipton, JAMA, 1998). Based on the results and selection of population is all medical students, I would assume stress and lack of rest is the main driver of the TTH. However, I feel the manuscript lacks a clearly summarized conclusion about the results.

Reply: We agree with the Reviewer's comment on the conclusions. We have modified and expressed the core of the study findings in the conclusion section.  

There are some minor issues:

  1. Format of the reference is not uniform.

Reply: It has been modified and revised all the references.

  1. Ref to ICHD-3 is not correct: International Classification of Headache Disorders-3 (beta version). Cephalalgia 2013;33:629–808.

Reply: It has been modified and revised all the references.

  1. The date range of data is not consistent in the abstract and methodology

Reply: It has been modified and revised as per the study grant that

  1. Avoid using language like "similar things."

Reply: It has been modified and revised.

Reviewer 2 Report

Dear authors,

The work is well done and arranged. I would suggest to speak about the possiblke causes of Headache (etiology) in the introduction, finding also references that talks about TMJ and its connection with headache. I would suggest to cite the neck pain and cite this (Saccomanno, S., Saran, S., Vanella, V., Mastrapasqua, R.F., Capogreco, M., Carretta, G., Pirino, A., Scoppa, F. Is there any correlation between malocclusion, temporomandibular disorders, and systemic disease? The importance of differential diagnosis (2022) Journal of Biological Regulators and Homeostatic Agents, 36 (2), pp. 149-156.).

Also the connection with orofacial pain is very important and stressful period of covid 19. Please look at this ( https://doi.org/10.3390/ijerph19127154 )

I would add something about the power of the study and specify the number of dental students and medical students.

FInally please connect the conclusions with results

Author Response

Thank you so much for your kind words about our research. We modified it as per your suggestion. Kindly review the article, and if any further needs to be improved, please give us a chance to change it as per the journal guidelines. 

Open Review

English language and style

( ) English very difficult to understand/incomprehensible
( ) Extensive editing of English language and style required
( ) Moderate English changes required
(x) English language and style are fine/minor spell check required
( ) I don't feel qualified to judge about the English language and style

Reply: The English were edited by one of the English language professional if any further is required, we will do it accordingly

Yes

Can be improved

Must be improved

Not applicable

Does the introduction provide sufficient background and include all relevant references?

( )

(x)

( )

( )

Are all the cited references relevant to the research?

( )

(x)

( )

( )

Is the research design appropriate?

(x)

( )

( )

( )

Are the methods adequately described?

( )

(x)

( )

( )

Are the results clearly presented?

( )

(x)

( )

( )

Are the conclusions supported by the results?

( )

(x)

( )

( )

Comments and Suggestions for Authors

Reply: The Introduction, Methods, results, conclusions, and references have been revamped and aligned with the latest available evidence.

Dear authors,

The work is well done and arranged. I would suggest to speak about the possiblke causes of Headache (etiology) in the introduction, finding also references that talks about TMJ and its connection with headache. I would suggest to cite the neck pain and cite this (Saccomanno, S., Saran, S., Vanella, V., Mastrapasqua, R.F., Capogreco, M., Carretta, G., Pirino, A., Scoppa, F. Is there any correlation between malocclusion, temporomandibular disorders, and systemic disease? The importance of differential diagnosis (2022) Journal of Biological Regulators and Homeostatic Agents, 36 (2), pp. 149-156.).

 Reply: We thank you for your encouraging words about our research work. And we have included this article as a reference for our article.  

Introduction has been rectified and modified as per the reviewers' suggestion

Also the connection with orofacial pain is very important and stressful period of covid 19. Please look at this ( https://doi.org/10.3390/ijerph19127154 )

 It has been included

I would add something about the power of the study and specify the number of dental students and medical students.

 It has been mentioned.

Finally please connect the conclusions with the results

It has been modified as per the reviewer's suggestions

Round 2

Reviewer 1 Report

Thanks for addressing my previous comments carefully, the overall quality and structure has been much improved.

There are several minor format issues I found in the corrected version, I will endorse it after those corrections:

1) Make sure the font is uniform.

2) The corrections of the references seem to be not appearing in the final text.

Reviewer 2 Report

the work is improved. The statistical analysis shoul be improved as said before with power and more details on the tests.